# Mechanisms of Spindle Positioning: Lessons from Worms and Mammalian Cells

**DOI:** 10.3390/biom9020080

**Published:** 2019-02-25

**Authors:** Sachin Kotak

**Affiliations:** Department of Microbiology and Cell Biology (MCB), Indian Institute of Science (IISc), Bangalore 560012, India; sachinkotak@iisc.ac.in; Tel.: +91-80-2293-2292

**Keywords:** mitosis, microtubules, spindle positioning, actin cytoskeleton, NuMA, dynein, myosin

## Abstract

Proper positioning of the mitotic spindle is fundamental for specifying the site for cleavage furrow, and thus regulates the appropriate sizes and accurate distribution of the cell fate determinants in the resulting daughter cells during development and in the stem cells. The past couple of years have witnessed tremendous work accomplished in the area of spindle positioning, and this has led to the emergence of a working model unravelling in-depth mechanistic insight of the underlying process orchestrating spindle positioning. It is evident now that the correct positioning of the mitotic spindle is not only guided by the chemical cues (protein–protein interactions) but also influenced by the physical nature of the cellular environment. In metazoans, the key players that regulate proper spindle positioning are the actin-rich cell cortex and associated proteins, the ternary complex (Gα/GPR-1/2/LIN-5 in *Caenorhabditis elegans*, Gαi/Pins/Mud in *Drosophila* and Gαi_1-3_/LGN/NuMA in humans), minus-end-directed motor protein dynein and the cortical machinery containing myosin. In this review, I will mainly discuss how the abovementioned components precisely and spatiotemporally regulate spindle positioning by sensing the physicochemical environment for execution of flawless mitosis.

## 1. Introduction

To efficiently grow and divide, all cells undergo a series of tightly regulated events known as the cell cycle. In eukaryotes, the cell cycle comprises Interphase and M-phase. M-phase is further divided into mitosis (or meiosis in germ cells) and cytokinesis. During mitosis, all animal cells establish an elegant diamond-shaped microtubule-based structure known as a mitotic spindle that is critical for ensuring error-free partitioning of the genomic, as well as intracellular contents (for details, please refer to [1,2]). In addition to this, the accurate positioning of the mitotic spindle is critical for the correct placement of the cleavage furrow, relative sizes and spatial organization of the daughter cells, and faithful segregation of the cell fate determinants during asymmetric cell divisions including in the stem cells (reviewed in [3,4,5,6,7,8,9]). In recent years, many studies in several cellular model systems have worked out mechanistic details employed by the cells to position the mitotic spindle in the three-dimensional cellular milieu. Notably, most of these mechanisms primarily rely on the dynamic astral microtubules that emanate from the centrosomes. These astral microtubules reach out either to the actin-rich cytoskeleton beneath the plasma membrane (referred to as cell cortex), interact with membranous organelles through minus-end-directed motor protein complex dynein to generate cytoplasmic pulling forces (for details please see the related review and work [10,11,12,13]), or interact with subcortical actin clouds that help in generating pulling force for accurate spindle positioning [14,15]. Depending on the cellular context, cells rely on one or all these mechanisms to accurately position the mitotic spindle. In addition to solely depend on the protein–protein-based mechanisms, emerging data suggest that cells also possess mechanisms whereby they rely on external mechanical signals to instruct the intracellular chemical environment to align the mitotic spindle correctly.

Here, I review the mechanisms of spindle positioning in *Caenorhabditis elegans* embryo and in mammalian cells. I will first discuss the evolutionarily conserved cellular machinery in *C. elegans* embryos and mammalian cells, followed by the discussion of how this intricate machinery spatiotemporally coordinate with mitotic progression to ensure proper spindle positioning. I encourage readers to glance at several excellent reviews that have highlighted the importance of upstream polarity regulators in guiding spindle positioning in *Drosophila* embryonic neuroblasts, *Drosophila* sensory organ progenitor cells (SOPs), *C. elegans* embryos, and mammalian epithelial cells in development, morphogenesis and stem cells [3,4,5,6,16,17,18,19,20,21]. I will further discuss the new paradigms whereby extrinsic physical and chemical signals are shown to modulate spindle positioning, and there I will also cover a few examples from the heterologous cellular models. I will then finish by alluding some interesting remaining questions; answering those will be helpful for better understanding the underlying mechanisms of spindle positioning in animal cells. 

## 2. Regulation of Spindle Positioning: Function of Key Players, Physical Environment, and Chemical Cues

### 2.1. The Ternary Complex and Associated Proteins: The Dynein Capturing Machinery at the Cell Cortex

In metazoans, proper positioning of the mitotic spindle is regulated by multiple means. However, one of the key pathways that regulates the proper positioning of the mitotic spindle in most cells examined is “cortical pulling”. This mechanism depends on specific sites on the cell cortex that capture and exert forces on astral microtubules. These forces then collectively act on the centrosomes that eventually position the mitotic spindle. Direct evidence for the pulling force generation in spindle positioning originated from the elegant spindle severing experiments with a UV-based laser microsurgery, whereby spindle severing lead to an outward movement of the centrosome towards the respective polar cell cortex [22,23,24]. Subsequent work revealed that the origin of such pulling force is the cell cortex [25,26]. How are astral microtubules captured at the specific cortical sites and thus help in generating pulling forces? Initial work in *C. elegans* one-cell embryo revealed that the pulling forces are primarily generated by an evolutionarily conserved ternary complex comprising a large coiled-coil protein (LIN-5), two almost identical tetratricopeptide (TPR) and GoLoCo domain-containing proteins (referred to as GPR-1/2 to represent a protein pair), and heterotrimeric G protein alpha subunits (GOA-1 and GPA-16 in *C. elegans*, collectively referred to as Gα from here onwards) (LIN-5/GPR-1/2/Gα) [27,28,29,30,31]. Because Gα is localized at the cell membrane through myristoylation, it anchors the entire ternary complex at the cell cortex beneath the plasma membrane [32,33]. In *C. elegans* one-cell embryo, spindle initially set up in the embryo center; however, under the control of intrinsic polarity regulators, the partition-defective proteins (PARs), it is displaced towards the posterior during late metaphase/early anaphase, and that results in an unequal division (Figure 1A) [4,34]. Loss of either LIN-5, GPR-1/2 or Gα resulting in the complete absence of the pulling forces, and an equal division of the one-cell embryo [28,29,30,31]. The apparent movement of the mitotic spindle at the posterior cortex is due to an asymmetric enrichment of the components of the ternary complex at the posterior cell cortex [28,29,30,35,36]. This data is in line with the previous assumption based on the centrosome disintegration experiment whereby it was calculated that there are approximately 50% more cortical force generator at the posterior cell cortex than at the anterior [25]. Analogous to the *C. elegans* embryos, the mitotic spindle in HeLa cells align in a stereotype axis when such cells are cultured either on uniform extracellular matrix (ECM) or grown on ECM-based micro-patterns [37,38,39]. For instance, when HeLa cells are cultured on the uniform fibronectin substrate, spindle align parallel to the substrate; however when such cells are cultured on the L-shape fibronectin-based micro-patterns, spindle align in the longest axis i.e., along the hypotenuse (Figure 1B) [37,38]. Notably, in HeLa cells, accurate spindle positioning either on uniform ECM or on ECM-based micro-patterns relies on the ternary complex consisting of NuMA/LGN/Gα_i1-3_ [33,40,41,42,43,44]. Similarly, spindle positioning in Madin-Darby Canine Kidney (MDCK) cells and keratinocytes during metaphase also depend on NuMA/LGN/Gα_i1-3_ [32,45,46,47].

How does the cortical anchoring of the ternary complex generate pulling forces? It is shown that LIN-5 (NuMA, Nuclear Mitotic Apparatus) directly associates with the minus-end-directed motor protein complex dynein and this interaction helps in the localization of dynein at the cell cortex [33,48,49,50,51,52]. It is believed the movement of cortically anchored dynein on the astral microtubules towards the centrosome generates pulling forces for proper spindle positioning (Figure 2). For efficient pulling in *C. elegans* one-cell embryo, it is necessary that astral microtubules retain dynamicity [48]. Otherwise, astral microtubules when hit the cell cortex may generate a counterforce in the opposite direction, and that may impede proper spindle positioning [48]. Surprisingly, in contrast to the *C. elegans* one-cell embryo, optogenetically targeted exogenous NuMA/dynein complex can efficiently pull taxol-stabilized microtubules in HeLa cells [51]. Therefore, it will be crucial to characterize whether endogenous levels of NuMA/dynein complex at the cell cortex in HeLa relies on the dynamic astral microtubules for efficient spindle pulling, and if not, what are the differences among the two systems. More direct evidence for the dynein-mediated pulling forces came from the elegant *in vitro*-based assay whereby yeast dynein anchored artificially to a micro fabricated surface is sufficient to capture microtubules, regulate microtubules dynamics and capable of generating pulling forces [53,54]. Interestingly, in *C. elegans* one-cell embryo, HeLa cells, and keratinocytes, dynein activation at the plus-end of the astral microtubules requires LIN-5/NuMA as an adaptor protein, and dynein alone at the cortex is not sufficient for efficient pulling force generation [51,52,55,56]. 

In HeLa cells, keratinocytes and other mammalian cell lines NuMA/dynein levels dramatically enrich at the cell cortex as the cells progress from metaphase to anaphase transition [47,57,58,59,60]. This NuMA-dependent cortical enrichment of dynein in anaphase is critical for efficient spindle elongation [58,61]. Surprisingly, in contrast to the one-cell *C. elegans* embryo, cortical enrichment of NuMA/dynein in mammalian cells during anaphase do not require Gα_i1-3_/LGN [43,47,59,60]. Furthermore, it is shown that in anaphase NuMA directly associates with membrane phosphoinositides, in particular, PtdIns(4)P (PI(4)P) and PtdIns(4,5)P_2_ (PI(4, 5)P_2_), and these lipid species are crucial for its membrane localization [43,60]. Analogous to mammalian cells in anaphase, in *Drosophila* epithelia, NuMA ortholog Mud localizes to the tricellular junctions (TCJs) independent of Pins and Gα pathway [62]. There as cells round up during mitotic entry TCJs localized Mud act as an interphase cell shape landmark for dynein localization and thus help in dictating spindle positioning in mitosis [62]. How does Mud localize to TCJs in *Drosophila* epithelia? Whether analogous to human cells, Mud requires specific phosphoinositides to localize at TCJs is not known, and future work will entail if membrane lipids or some other yet unknown anchor for Mud is needed for its localization at TCJs to coordinating cell geometry with the division axis.

Are there more proteins involved than these simplistic ternary complex components to influence spindle positioning through cortical pulling? Multiple proteins have been implicated in the correct localization of the components of the ternary complex. For example, in *C. elegans* one-cell embryos, a DEP domain-containing protein LET-99 localizes at the posterior-lateral region of the cell cortex and helps in the asymmetric distribution of the LIN-5/GPR-1/2 for the proper pulling force generation [63,64]. Similarly, in the one-cell embryos, Gβ GBP-1 is needed for the proper positioning of the mitotic spindle [65,66]. Gβ competes for Gα.GDP for GPR-1/2 binding and thus negatively regulates pulling forces [67]. Notably, in mammalian epithelial cell culture and mouse retinal progenitor cells LGN interacting protein SAPCD2 is needed for spindle positioning. SAPCD2 interacts with Gα.GDP, and negatively regulates cortical localization of LGN and therefore influences spindle positioning [68]. Moreover, a cofactor of the p97 AAA ATPase p37/UBXN2B regulates spindle positioning in mammalian cells by controlling the localization of cortical NuMA [69,70]. Furthermore, a conserved adherens junction structure and a cell polarity regulator Dlg1 is crucial for LGN recruitment for accurate spindle positioning in HeLa cells as well as in chicken neuroepithelium [71]. Since the ternary complex and dynein act at the cell cortex in the proximity of the actin-rich cytoskeleton, it is assumed that proteins that help in maintaining actin cytoskeleton would directly or indirectly influence the localization of the ternary complex components/dynein. Indeed, several recent reports have highlighted the involvement of numerous actin-associated proteins in choreographing the localization/activity of the ternary complex. In this realm, 4.1 families of proteins are needed for the stabilization of NuMA at the cell cortex in HeLa cells and keratinocytes [43,47], and a junctional protein Afadin is shown to interacts with F-actin and LGN and required for proper spindle positioning during epithelial morphogenesis [72]. Also, an actin-associated protein MISP is critical for proper spindle positioning by regulating cortical distribution of NuMA and dynein-dynactin complex in HeLa cells [73,74]. Similarly, E-Cadherin is shown to directly interact with LGN and helps in directing assembly of NuMA/LGN complex at the cell-cell contacts so that cell division axis can be coupled with the intracellular adhesion [75]. From these data, it also appears that the link between the components of the ternary components and actin/E-Cadherin function on spindle positioning would be relevant in the context of a tissue, where cells need an extrinsic mechanism to communicate with each other for maintaining proper tissue architecture. 

### 2.2. Linking Extrinsic Mechanical Forces to Spindle Positioning

Mammalian cells become round as they enter mitosis; however, they remain connected to the underlying substrate through retraction fibers (membranous tubes containing thick actin filaments). Elegant work using HeLa cells grown on the fibronectin-based micro-patterns revealed that when these cells are cultured on different shapes of micro-patterns, cells adapt to these shapes (Figure 1B) [38]. However, as these cells enter metaphase, the mitotic spindle aligns along the longest axis determined by the interphase geometry of these micro-patterns (Figure 1B) [38]. These observations suggested that cells keep the memory of the interphase cell shape that guide spindle positioning in mitosis. Importantly, these observations also imply that the external environment is crucial in regulating the positioning of the mitotic spindle. Because of the good correlation between organization of the retraction fibers and the geometry of the micro-patterns and spindle positioning, it was assumed that the retraction fibers could relay mechanical force from outside to the inside of the cell that eventually regulate spindle positioning [38,39]. To test the function of adhesion fibers-mediated external mechanical forces on mitotic spindle positioning, Fink et al. performed ingenious laser-based microsurgery experiments to ablate retraction fibers in HeLa cells cultured on the micro-patterns. These set of experiments revealed that retraction fibers are essential for guiding the proper spindle positioning, not only during G2 to M transition but also while the cells are in metaphase [14]. Furthermore, by conducting uni-axial stretch to the cultured HeLa cells and keratinocytes during mitosis, it was shown that the spindle aligns towards the applied stretch, further suggesting that external force can induce spindle positioning during mitosis [14,47]. How external forces regulated by retraction fibers polarize the cells for proper spindle positioning remain incompletely understood. Intriguingly, external forces governed by retraction fibers rely on internal astral microtubules to accurately position the mitotic spindle, suggesting external information is getting relayed inside the cells [14,38]. Notably, cytoplasmic actin cloud was proposed to control spindle positioning in response to the retraction fibers [14]. Subsequent work revealed that an unconventional myosin motor Myo10 is required to couple extracellular forces with the intracellular actin cloud [15]. The ternary complex components are also needed for proper spindle positioning when HeLa cells are grown on fibronectin-based micro-patterns [42,43,44], and importantly, there is a strong correlation between the cortical localization of the ternary complex and dynein with the organization of retraction fibers during metaphase [40,76]. Moreover, cortical dynein adaptor NuMA is the critical in aligning the mitotic spindle in mammalian keratinocytes upon uni-axial stretch [47]. Overall, these observations indicate that the ternary complex components are also involved in sensing the external mechanical forces. These data altogether suggest that NuMA/dynein pathway work in combination with the actin cloud/myosin pathway in orchestrating mitotic spindle behavior in HeLa cells. Furthermore, Ezrin/radixin/moesin (ERM) proteins that are the key regulators for the cortical F-actin organization are shown to play a crucial function for polarized localization of LGN and NuMA in HeLa cells grown on the micro-patterns [76]. However, the identity of the central molecule/s of the ternary complex that senses the external force, and more importantly what is the relationship between intracellular actin cloud, ERM proteins, and the ternary complex components remain incompletely understood. 

Interestingly, a relatively recent study by Sugioka and Bowerman has challenged the unifying theme of the involvement of the ternary complex components and dynein in orchestrating spindle positioning in the *C. elegans* embryos. They have uncovered an entirely new paradigm for controlling spindle positioning. This mechanism depends on the external physical contact to position the mitotic spindle in the anterior blastomere AB of the two-cell stage of *C. elegans* embryo (Figure 3A,B) [77]. As reported previously, the authors showed that GPR-1/2/dynein pathway is dispensable for the spindle positioning in the AB cells, and intriguingly, astral microtubules are also not required for defining the division axis in AB blastomere [30,77]. Furthermore, the authors neatly showed that physical contact either by P_1_ cell or by carboxylate-modified polystyrene bead is sufficient to define the spindle axis in the AB blastomere, suggesting that external mechanical cues are the key for the proper spindle positioning (Figure 3C) [77]. How does cell contact-mediated information regulate spindle positioning in the AB cell? Here the authors revealed that extrinsic mechanical anisotropy induces cortical myosin II flow that is crucial for generating forces to align the mitotic spindle. Such anisotropy in the cortical myosin flow has earlier been proposed in the positioning of ABa and ABp division axis [78]. Therefore, it would be important to investigate the molecular mechanisms by which physical inputs instruct myosin flow to control spindle dynamics. 

### 2.3. Extrinsic Chemical Code for Guiding Positioning of the Mitotic Spindle 

Proper positioning of the mitotic spindle in the one-cell of *C. elegans* embryo is dependent on the intrinsic cell polarity-based mechanisms regulated by partition-defective proteins (PARs) [20]. At the four-cell stage of *C. elegans* embryo, EMS blastomere undergoes asymmetric division along the anterior-posterior (A-P) axis to form a posterior daughter E cell that produces endoderm, and an anterior MS cell that forms mesoderm (Figure 3D–F) [79,80,81,82]. Alignment of the mitotic spindle along the A-P axis in EMS cell requires the 90° rotation of centrosome-nucleus complex before the nuclear envelope breakdown (Figure 3A,B) [83]. Beautiful experiments using blastomeres isolated from wild-type embryos demonstrated that positioning of the centrosomes and asymmetric division of the EMS cell require signaling from the posterior P_2_ blastomere [80,84]. If P_2_ is not kept next to EMS blastomere, the mitotic spindle elongates at the axis established by the centrosome migration, but no rotation of centrosome-nucleus complex takes place [84]. Therefore, it was suggested that contact of P_2_ is crucial for the positioning of centrosome-nucleus complex. Notably, mutations in the components of the Wnt-signaling pathways for instance Wnt ligand (MOM-2 in *C. elegans*), the Frizzled receptor (MOM-5), and Disheveled adaptors (MIG-5 and DSH-2), play key functions for the spindle positioning in EMS cell [85,86]. Loss of these factors cause mis-positioning of the spindle in EMS cell and that impact specification of the germ cell layers [85,87]. The role of Wnt on spindle positioning in EMS cell is independent of transcriptional activation, which is occasionally linked with canonical Wnt signaling [85]. Since microtubules are essential, and gene-regulation is not needed for such rotations, it was assumed that P_2_-mediated Wnt signaling directly impact the cytoskeleton for positioning of the centrosome-nucleus complex [84,85,88]. To properly position the mitotic spindle, the Wnt-signaling pathway genetically interacts with the components of the ternary complex and associated proteins LIN-5 and LET-99 [63,89]. By using a combination of Wnt deficient P_2_ cell or a polystyrene bead, recent work from Sugioka and Bowerman has suggested that this Wnt-pathway not only induces LIN-5/dynein-dependent spindle positioning in the EMS cell, but also inhibits cortical actomyosin to orchestrate the division axis in the EMS cell [77]. However, despite the requirement of LIN-5 and dynein for spindle positioning in EMS cell [89], LIN-5 and dynein are not asymmetrically distributed at the cell cortex [90,91]. This raises an interesting possibility that either the ternary complex components are asymmetrically activated in such a cellular system or could act in concert with other components of the Wnt signaling to control spindle positioning in EMS cell. Importantly, similar to the *C. elegans* embryo, immobilized Wnt protein on beads can act as a spatial cue to direct accurate positioning of the mitotic spindle in mouse embryonic stem cells [92]. This indicates that the Wnt-based chemical cues guiding the position of the mitotic spindle could be evolutionarily conserved. Therefore, it would be quite fascinating to explore how Wnt signaling mechanistically controls the cytoskeleton to direct proper positioning of the mitotic spindle. 

Taken together, it is becoming increasingly apparent that various cellular systems use multiple mechanisms whereby force generating machinery coordinate with the physical and chemical cues to position the mitotic spindle. These mechanisms may work in concert, or perhaps in an antagonistic way to modulate the spindle dynamics in time and space. 

## 3. Spatiotemporal Control of Spindle Positioning

Another key aspect of pulling force generation is that such forces should act only after the onset of mitotic spindle assembly; otherwise, premature excess forces may perturb mitotic spindle assembly and may also affect accurate spindle positioning. In *C. elegans* one-cell embryos, the ternary complex components GPR-1/2 or LIN-5 are uniformly distributed at the cell cortex in metaphase, and only during metaphase to anaphase transition one can appreciate the cortical enrichment of LIN-5/GPR-1/2 at the posterior cortex that coincides with the spindle movements [28,29,35,36]. In an elegant study using *C. elegans* one-cell embryos overexpressing GPR-1 it was revealed that premature excess pulling forces before bipolar spindle assembly cause formation of two monopolar spindles containing chromosomes from either the maternal or the paternal pronucleus eventually resulting in the segregation of the maternal or the paternal genome in two different daughters [93]. Also, excess pulling forces during metaphase or anaphase may impact proper spindle positioning [33], therefore, it is necessary that the regulators of spindle positioning are tightly and dynamically regulated during mitosis. Data coming from several metazoan cellular systems collectively suggest that posttranslational modifications, in particular phosphorylation of the components of the ternary complex, play a key function in this regulation. In *C. elegans* embryo, the anterior PAR complex component aPKC phosphorylate LIN-5 at the anterior and inhibits cortical pulling forces generation at the anterior, and thus, the collective net force is higher on the posterior centrosome in the one-cell embryo [94]. Interestingly, in mammalian epithelial cells, LGN was shown to be negatively regulated by aPKC. In this case, aPKC was shown to phosphorylate LGN, and this causes the binding of 14-3-3 and unbinding of LGN from the apical cortical region [95]. In the absence of aPKC, LGN stays at the apical cortex and this impacts planer spindle positioning. In HeLa cells, centrosomal Polo-like kinase 1 (Plk1) gradient was implicated in negatively regulating cortical dynein localization in metaphase [42]. Later, it was revealed that centrosomal Plk1 impact cortical dynein either through LGN or by regulating a protein that acts upstream of LGN [96]. However, recently NuMA was identified as a direct target of Plk1, and the effect on LGN cortical localization by Plk1 inhibition could stem from the fact that both LGN and NuMA are co-dependent on each other in metaphase for their cortical enrichment [97]. Plk1 phosphorylates NuMA at SS1833/34, and it is shown that this phosphorylation regulates the cortical distribution of NuMA and thus dynein for proper spindle positioning [97,98]. In addition to Plk1, several other mitotic kinases such as Aurora A, and Cdk1/cyclin B (referred to as Cdk1 afterward) are also implicated in the regulation of spindle positioning through NuMA. Upon Aurora A inactivation or its depletion by RNAi in HeLa cells, NuMA fails to localize at the cell cortex, and that impact proper spindle positioning [99,100]. It is further revealed that the inability of NuMA to localize at the cell cortex upon Aurora A inactivation is due to change in its dynamic behavior. Importantly, Aurora A-mediated phosphorylation of Serine 1969 (S1969) is implicated in modulating dynamic behavior of NuMA and influencing its cortical localization [99]. Similarly, Cdk1 phosphorylates NuMA in metaphase at Threonine 2055 and negatively regulates its cortical localization [47,58,59]. Intriguingly, only non-phosphorylated NuMA species at T2055 localizes at the cell cortex in metaphase, suggesting that other than kinases, phosphatases could play a substantial role in fine-tuning the localization of the components of the ternary complex for proper spindle positioning [58]. Other than the involvement of the mitotic kinases in the regulation of spindle positioning in human cells, Abelson kinase (ABL1) was identified in a kinase-targeted RNAi-based screen in HeLa, and it was shown to phosphorylate NuMA at the conserved residue Tyrosine 1774 (Y1774), and this phosphorylation impacts the cortical distribution of LGN/NuMA [101]. It is notable that in most of the instances for the phospho-regulation of the ternary complex, the molecular target is NuMA, why is that? One simple possibility is that we have not yet extensively analyzed the phosphorylation status of the other components of the ternary complex such as Gα or LGN (GPR-1/2). Alternatively, it could well be that since unlike Gα/LGN(GPR-1/2), NuMA regulates cortical localization of dynein both in metaphase and in anaphase, evolution has chosen NuMA as a strategic molecule to phospho-modulate its localization at the district phases of mitosis. 

## 4. Conclusions

The inherent ability of the mitotic spindle to position in the particular reference axis is vital to generate cell fate diversity and to ensure that cells are organized in a proper three-dimensional arrangement within a tissue. In the last few years we have learned the existence of multiple players that regulate correct positioning of the mitotic spindle in time and space. However, despite the discovery of several sophisticated building blocks that organize and spatiotemporally control spindle positioning, our understanding how these individual pieces collectively communicate to regulate spindle positioning is still far from clear. For instance, it is becoming more apparent that cortical actin and its associated proteins play a crucial role of spindle positioning, but whether these players directly or indirectly affect spindle positioning by modulating the localization/activity of the ternary complex and dynein remains elusive. Also, it would be exciting to explore the biophysical and structural features of the ternary complex and associated proteins to obtain some novel insights. For instance, it would be important to dissect the structure details of NuMA (LIN-5) and to test whether its property of being large coiled-coil assist in keeping dynein at the outermost edge of the thick actin cytoskeleton for efficient capturing of the astral microtubules. It would be equally important to test if analogous to mammalian cells in culture, whether various mammalian tissues also employ multiple mitotic kinases to fine-tune accurate positioning of their spindle in a spatiotemporal manner. In this realm whether mitotic phosphatases also play a key role in the modulating the levels/activities of cortical force generators would be an important question for future studies. Despite the apparent role of mechanical processes in regulating spindle positioning, how mechanical cues are guiding chemical machinery to orient mitotic spindle is not yet very much evident, and thus new data in this theme will definitely strengthen our current understanding. There have been good correlations between spindle positioning defects with human diseases including tumorigenesis and neuropathological disorders [102,103,104,105,106], and so one crucial aim of the future work would be to discover if there is a causative association between such human pathology and spindle positioning. Overall, with the advancement of a myriad of state-of-the-art technologies such as optogenetics [51,52], and acute inactivation/degradation of proteins using chemical probes, exciting time lie ahead of us where we can expect novel insights in the theme of spatial and temporal regulation of spindle positioning for error-free cell division. 

## Figures and Tables

**Figure 1 biomolecules-09-00080-f001:**
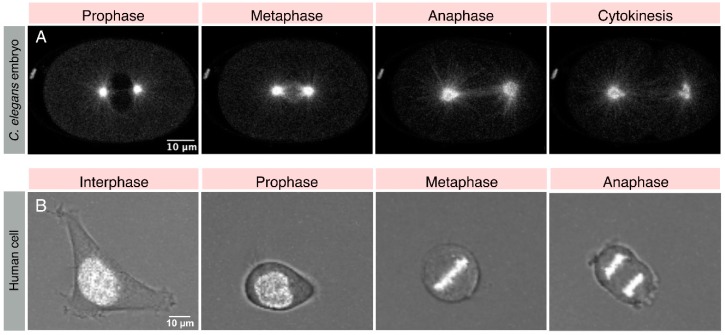
Spindle positioning in *Caenorhabditis elegans* embryos and human cells. Images from the live-recording of the one-cell stage of *C. elegans* embryo expressing mCherry-tubulin (in grey) to label the microtubules are at various stages of cell cycle (**A**). Please note asymmetric spindle positioning in anaphase cells along the anterior-posterior axis, which would eventually lead to the unequal division of the one-cell embryo (not shown). Images from live-recording of HeLa cells stably expressing mCherry-H2B (in grey) to label the chromatin in interphase, and mitotic chromosomes at various stages of mitosis cultured on fibronectin-based L-shape micro-pattern. Please note that HeLa cell adapts the shape of the pattern during interphase; however, during metaphase when the cell is round spindle aligns along the longest axis. This causes chromosomes to separate along the longest axis when spindle elongates during anaphase (**B**).

**Figure 2 biomolecules-09-00080-f002:**
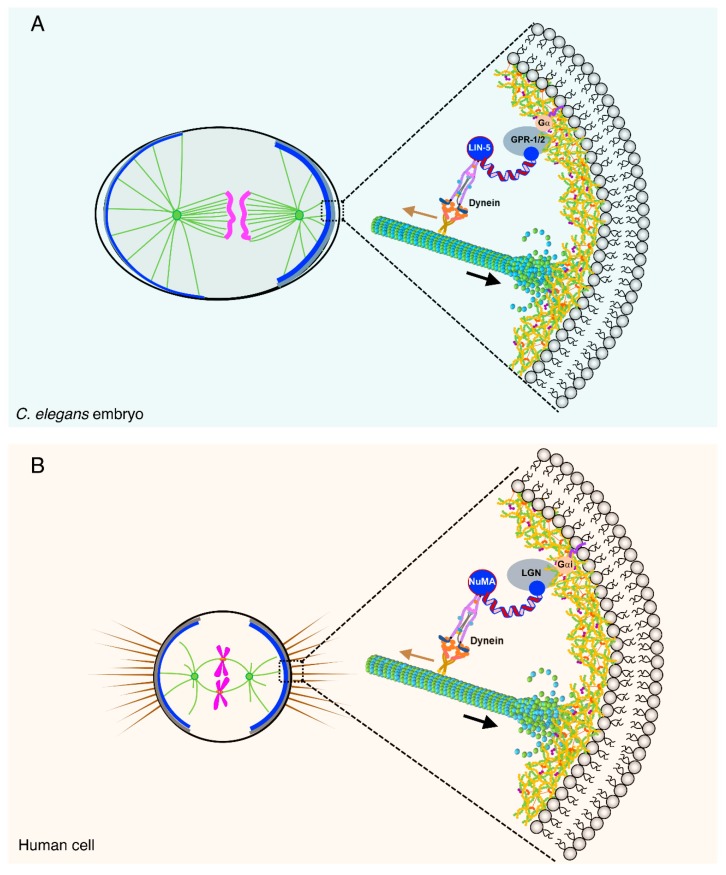
Cortically anchored ternary complex control spindle positioning in animal cells. Working model of spindle positioning in *C. elegans* embryo (**A**) and human cells (**B**). The ternary complex (LIN-5/GPR-1/2/Gα in *C. elegans* and NuMA/LGN/Gαi) is anchored below the cell cortex and recruits dynein motor protein complex. Dynein being a minus-end-directed microtubule-dependent motor complex attempts to move towards the centrosomes (shown by light brown arrow), but since it is anchored it instead pulls the astral microtubules resulting in the generation of the pulling forces towards the cell cortex (shown in black arrow). Please see text for more details.

**Figure 3 biomolecules-09-00080-f003:**
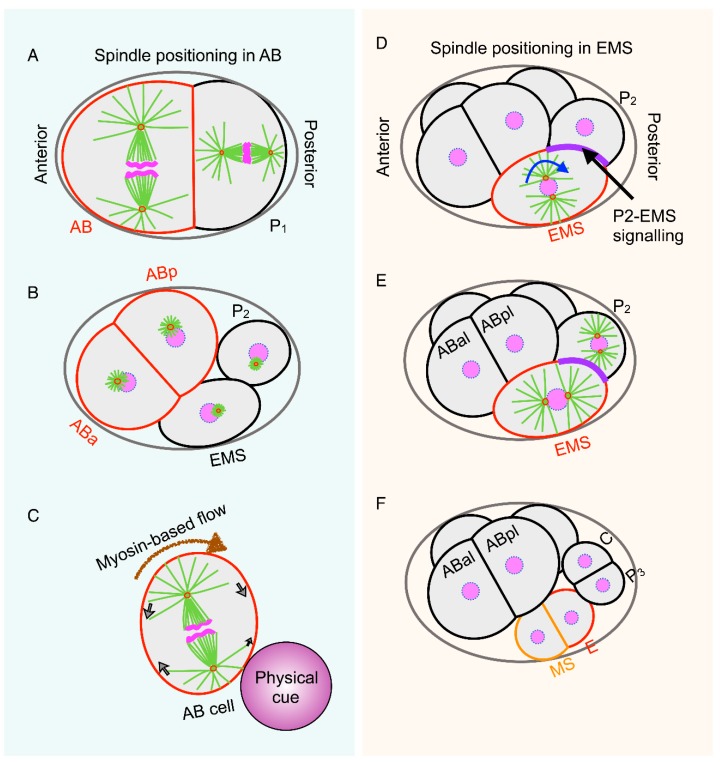
Spindle positioning mechanisms in AB and EMS cells of *C. elegans* embryo. The mitotic spindle in AB and P_1_ cells at the two-cell stage of *C. elegans* embryo aligns parallel and perpendicular to the to the plane of AB-P_1_ cell contact respectively (**A**,**B**). Importantly, the alignment of the mitotic spindle and the cleavage plane in the AB cell is independent to GRP-1/2/dynein, and microtubules, but rather dependent on the unequal myosin-driven cortical flow that generates force to specify the position of the cleavage plane. Importantly, contact with an inert bead (shown in magenta) is sufficient to create the anisotropy in the myosin flow, and this creates an imbalance in the actomyosin-driven forces which defines the division site (**C**). Arrows represent the myosin flow, whereby it is slowest in the region where the AB cell contacts the bead (physical cue). Asymmetric division in EMS cell (shown on the right) into MS and E cell (**D**–**F**). Prior to division, centrosome-nucleus complex rotates 90° under the influence of the Wnt signaling (shown in magenta) from the P_2_ cell (**D**,**E**). Wnt-signaling acts in concert with LIN-5/dynein pathway to instruct the mitotic spindle to align along the anterior-posterior axis. Please see text for more details.

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
