# Peer review of "Mechanisms of Spindle Positioning: Lessons from Worms and Mammalian Cells"

_biomolecules, 2019, doi:10.3390/biom9020080_

Round 1

Reviewer 1 Report

Kotak has written a thorough review of the role of the ternary complex of Galpha/GPR/NUMA and its recruitment of dynein to regulate mitotic spindle positioning. This topic is covered well, bridging work in different model systems very nicely. However, I do have two criticisms as to how the abstract sets up the review, and what the review actually covers. 

First, the review is focused entirely on spindle positioning and not on how the spindle influences the positioning of the cleavage furrow during cytokinesis. However, the abstract emphasizes the role of the spindle in specifying the cleavage furrow, while the review does not really address the relationship between spindle positioning and cleavage furrow positioning. I think it would be better to simply focus on spindle positioning and de-emphasize how the spindle positions the cleavage furrow. This relationship can of course be mentioned, but it is overly emphasized in the abstract. 

Second, the review abstract refers to both the role of dynein and the role of the cortical actomyosin cytoskeleton in the regulation of spindle positioning. But the focus of the review is almost entirely on the role of dynein and the NUMA complex that recruits dynein. While some research on the role of the cortical actomyosin cytoskeleton is covered, the abstract makes it seem as if both inputs will receive equal attention, when really the focus is on NUMA and dynein. Again, framing the review as focusing on the role of NUMA/dynein would make more sense. Along these lines, it is curious that the review references Sugioka and Bowerman (2018) in Developmental Cell, with respect to the influence of Wnt signaling on spindle orientation at the 4-cell stage in C. elegans, without ever mentioning that those authors argue that cortical actomyosin dynamics can orient the axis of cell division independently of the NUMA/dynein mechanism. Moreover, those authors propose that Wnt signaling acts by opposing this cortical actomyosin mechanism, such that a dynein-dependent mechanism might then orient the mitotic spindle in the 4-cell stage blastomere EMS. Kotek cites this manuscript in the context of EMS division without ever mentioning the authors’ hypothesis that Wnt signaling counteracts the cortical actomyosin mechanism that orients the axis of division independently of NUMA and dynein.

I recommend that the author should either clearly state that this review focuses on the role of NUMA and dynein in spindle orientation, or provide a more thorough account of (i) how spindle positioning influences furrow positioning, and of (ii) how cortical actomyosin might influence the axis of cell division independently of the NUMA/dynein.

Finally, there are numerous English grammar errors throughout the manuscript, mostly related to proper agreement of noun and verb with respect to singular versus plural nouns. 

Author Response

This reviewer mentioned that “Kotak has written a thorough review of the role of the ternary complex of Galpha/GPR/NUMA and its recruitment of dynein to regulate mitotic spindle positioning. This topic is covered well, bridging work in different model systems very nicely”.

However, had two major criticisms, that I have tried to address in the revised manuscript as mentioned below.

Major Concerns:

1.    First, the review is focused entirely on spindle positioning and not on how the spindle influences the positioning of the cleavage furrow during cytokinesis. However, the abstract emphasizes the role of the spindle in specifying the cleavage furrow, while the review does not really address the relationship between spindle positioning and cleavage furrow positioning. I think it would be better to simply focus on spindle positioning and de-emphasize how the spindle positions the cleavage furrow. This relationship can, of course, be mentioned, but it is overly emphasized in the abstract.

Response. I want to sincerely thank the reviewer for her/his in-depth suggestions on this review article. I am in agreement with the reviewer that this review is mainly on spindle positioning and not on how the position of the spindle influences cleavage furrow during cytokinesis. The reviewer will notice that I have modified the title to focus mainly on the spindle positioning, and slightly modified the abstract, thus not to overemphasize how the spindle positioning governs the placement of the cleavage furrow.

2.    Second, the review abstract refers to both the role of dynein and the role of the cortical actomyosin cytoskeleton in the regulation of spindle positioning. But the focus of the review is almost entirely on the role of dynein and the NUMA complex that recruits dynein. While some research on the role of the cortical actomyosin cytoskeleton is covered, the abstract makes it seem as if both inputs will receive equal attention, when really the focus is on NUMA and dynein. Again, framing the review as focusing on the role of NUMA/dynein would make more sense.

Response. I thank the reviewer for her/his suggestions. The reviewer will notice that I have tried to extend the discussion on the mechanisms of spindle positioning beyond what is regulated by the ternary complex by giving some more examples whereby actin-cytoskeleton associated proteins are shown to play an essential function for proper spindle positioning. The second and third reviewer also suggested to include more reference related with the role of actin cytoskeleton/associated proteins in spindle positioning. See the related points #1 and #3 of the second and third reviewers respectively.  

3.    Along these lines, it is curious that the review references Sugioka and Bowerman (2018) in Developmental Cell, with respect to the influence of Wnt signaling on spindle orientation at the 4-cell stage in C. elegans, without ever mentioning that those authors argue that cortical actomyosin dynamics can orient the axis of cell division independently of the NUMA/dynein mechanism.

Response. I apologize for not being sufficiently clear in referring to Sugioka and Bowerman (2018) with respect to NuMA/dynein independent mechanisms of spindle positioning. In the revised manuscript, in the section ‘Linking extrinsic mechanical forces to spindle positioning’ on p.11-12, I have now added a whole new section on the spindle positioning in AB blastomere. I have also included a new Figure 3A-C to emphasize the role of myosin-driven cortical flow independent of astral microtubules in spindle positioning in the AB cell.

4.    Moreover, those authors propose that Wnt signaling acts by opposing this cortical actomyosin mechanism, such that a dynein-dependent mechanism might then orient the mitotic spindle in the 4-cell stage blastomere EMS. Kotek cites this manuscript in the context of EMS division without ever mentioning the authors’ hypothesis that Wnt signaling counteracts the cortical actomyosin mechanism that orients the axis of division independently of NUMA and dynein.

Response. I further thank the reviewer for her/his critique. As also mentioned by the third reviewer (major concern #7), I have tried to substantially extend the discussion on the role of Wnt-based signalling in spindle positioning. This is covered in section ‘Extrinsic chemical code in guiding positioning of the mitotic spindle’ on page 12-13. The reviewer will see that I have specifically tried to include the past/latest literature of how we view the positioning of the spindle is regulated in the EMS cell/mouse embryonic stem cells. Also, as suggested by the reviewer, I have included on p. 13 the hypothesis of Sugioka and Bowerman that a Wnt-pathway not only induce LIN-5/dynein dependent spindle positioning but also inhibits cortical myosin to orchestrate division axis in the EMS cell.

5.    I recommend that the author should either clearly state that this review focuses on the role of NUMA and dynein in spindle orientation, or provide a more thorough account of (i) how spindle positioning influences furrow positioning, and of (ii) how cortical actomyosin might influence the axis of cell division independently of the NUMA/dynein. Finally, there are numerous English grammar errors throughout the manuscript, mostly related to proper agreement of noun and verb with respect to singular versus plural nouns.

Response. As mentioned in response to major point 2, I have added now more insight on the Gα/LGN/NuMA independent mechanisms of regulating spindle positioning. In doing so, at some instances, I have also covered Gα/LGN independent mechanisms of spindle positioning in anaphase cells as suggested by the third reviewer (major concern #6). Moreover, I have further included a few more examples from cellular systems such as Drosophila epithelia. In the revised version, I tried to rectify grammar errors related to the proper agreement of noun and verb concerning singular versus plural nouns. With this, I am hoping that the reviewer will find revised manuscript a useful asset for investigators working in the field of spindle positioning.

Reviewer 2 Report

The review manuscript “Mitotic spindle positioning: a process of collective team effort to determine the furrow initiation site” by Dr. Kotak reports the current knowledge on molecular mechanisms underlying the process of mitotic spindle orientation, with emphasis on C. elegans embryos and vertebrate cells. More specifically, the author describes the working principles of the conserved NuMA/LGN/Galphai complex that localises at the cell cortex and captures astral microtubules by recruiting dynein, and on the ancillary proteins regulating the localisation of NuMA and LGN during mitotic progression. The second part of the manuscript describes the extracellular signals instructing spindle orientation as observed in cells on dividing micro patterns or near wnt3a-coated beads. The post-translational modifications coordinating spindle orientation with cell cycle are also summarized.

Overall the manuscript is well designed and written, and provides an exhaustive survey on the literature available on an interesting topic. Therefore, I am in favour of publication on Biomolecules, provided that a few minor issues are addressed, as outlined below:

1. In describing the proteins implicated in NuMA/LGN/Galphai cortical recruitment I would include p37/UBXN2B (Lee at al, JCB 2018), MISP (I. Hoffmann lab) and Dlg1 (Saadaoui et al, JCB 2014).

2. In reporting on the various phosphorylations and proteins contributing to spindle orientation, I would make more explicit the cellular system and the mitotic phase when these events/cellular components are active.

Typos:

- pg 3 line 70: interacts should be interact

- pg 3 line 74: a reference on acting clouds is missing here (though it is present later in the text)

- pg 4 line 95-96: I think this sentence needs some rephrasing 

- pg 5 line 113: the references reported refer to vertebrate cells, while papers on C. elegans would be more appropriate here, if available

Author Response

This reviewer mentioned that “manuscript is well designed and written, and provides an exhaustive survey on the literature available on an interesting topic” but requested that we address few minor issues, which I have done as explained below.

Concerns:

1.    In describing the proteins implicated in NuMA/LGN/Galphai cortical recruitment I would include p37/UBXN2B (Lee at al, JCB 2018), MISP (I. Hoffmann lab) and Dlg1 (Saadaoui et al, JCB 2014).

Response. I thank the reviewer for these suggestions. I have now explicitly mentioned these findings in the text on p. 8.

2.    In reporting on the various phosphorylations and proteins contributing to spindle orientation, I would make more explicit the cellular system and the mitotic phase when these events/cellular components are active.

Response. Thanks –corrected.

3.         - pg 3 line 70: interacts should be interact

Response. Thanks –corrected.

4.         - pg 3 line 74: a reference on acting clouds is missing here (though it is present later in the text)

Response. Corrected as well.

5.         - pg 4 line 95-96: I think this sentence needs some rephrasing

Response. The sentence is rephrased, and I hope it reads better now.

6.         - pg 5 line 113: the references reported refer to vertebrate cells, while papers on C. elegans would be more appropriate here, if available

Response. I thank the reviewer for the suggestions. In an earlier work (Kotak et al., 2012 JCB), we have shown that myristoylation of Ga is critical for the spindle positioning in the one-cell embryo, and Ian Macara’s lab has previously revealed (Du and Macara, 2004 Cell) that cortical anchoring of Ga is key for proper targeting of the components of the ternary complex in mammalian cells, and thus the choice of the references seem appropriate here.

Reviewer 3 Report

In this Review, Kotak focuses on the molecular mechanisms of spindle positioning and orientation. The author provides a comprehensive view of the pioneer literature describing the main molecular complex at the core of spindle orientation. This core complex is composed by Gai, LGN and NuMA (hereafter called the “ternary complex”). In addition, the author discusses novel insights on how this molecular complex works, published during the last 5 years. The author focuses on C.elegans and cultured human cells, allowing a deep discussion of the literature available in these two models. Another interesting point of this review is that the author considers not only the molecular regulation of the ternary complex, but also its interaction with the actin cytoskeleton, and its modulation by external forces.

I suggest publication of this review once the following points are addressed: 

Major points

  1) It would be important that the author justifies the choice of focusing on C.elegans and cultured cells, mentioning for instance the advantages of the models and/or the resemblances between them. Also, the author should mention briefly other main models in which the core regulators were first described (Drosophila neuroblasts, for instance).

2) There is a lack of description of the different models of spindle orientation or positioning in  cultured cells: spindle orientation with respect to the substratum/ spindle orientation guided by micropatterns/ spindle positioning along its own axis. The mechanisms acting in each type of orientation might be different, so the author should describe better these different models, and specify throughout the review in which model each mechanism was found. Similarly, the explanation of the micropattern based system is incomplete (lines 128-129).

3)      The author could go deeper in the discussion on the interaction of the ternary complex with actin related proteins and the actin cortex, by mentioning more example proteins or other mechanisms by which the actin cortex may regulate spindle orientation (e.g. Machicoane et al., JCB 2014; Castanon et al., NCB 2013). What do we know about the contribution of the actomyosin cytoskeleton to spindle positioning in C.elegans? (e.g:  Redemann et al., Plos One 2010, Sugioka and Bowerman, Dev Cell 2018).

4)      In lines 195 to 198, the author suggests a link between the ternary complex, cellular junctions and actin, and its function in tissues. There are novel insights into this subject, so some references should be at least included (e. g. Bosveld et al., Nature 2016).

5)      In the Section « Coupling extrinsic mechanical forces to instructing spindle positioning »:

While the author indicates that the link between retraction fibers, actin clouds and the ternary complex is not clear (lines 220 to 222), the articles by Machicoane et al. (JCB, 2014) and Tame et al. (Cell Cycle, 2014) link the localization of the ternary complex with the distribution of retraction fibers. Besides, the article by Kwon et al. (Dev Cell, 2015) strongly suggests that the LGN-dynein pathway acts in parallel to the actin cloud-myosin 10 pathway.

6)      In section « Spatiotemporal control of spindle positioning »:

-          Plk1 was suggested to regulate both dynein and LGN cortical localization in separate studies (Kiyomitsu and Cheeseman, NCB 2012; Tame et al., EMBO reports, 2016); the author could discuss both datasets.

-           A temporal description of the biochemical/localization events differing between metaphase and anaphase is lacking, in particular with respect to NuMA phosphorylation states and its cortical receptors (Kiyomitsu and Cheeseman, Cell 2013; Kotak et al., EMBO J 2013; Kotak et al., EMBO J 2014).

7)      The choice of dedicating a section to the role of Wnt in spindle orientation is unclear, as this section does not go deep enough in the subject and seems a bit disconnected to the rest of the review. I suggest to either remove this section, or extend its content.

Minor points

1)      I suggest to better define spindle orientation and spindle positioning at the beginning of the review.

2)      In lines 67-68 the author states that all spindle orientation mechanisms rely on astral microtubules. There are some exceptions as shown by Lazaro Dieguez et al. (Mol Biol of the Cell, 2015).

3)      The author mentions that the regulation of the spindle orientation ensures « error-free » cell division, or « flawless » division. However, spindle orientation per se does not ensure error free division neither spindle misorientation induces defective mitosis (concerning genomic integrity at the end of division, if this is what the author meant).

           4)      In line 189, a reference to Kiyomitsu and Cheeseman (Cell, 2013) should be included.

             5)    In conclusion section: In line 342, a reference to articles using optogenetics to study spindle orientation should be included: Fielmich et al. (Elife, 2018) and Okumura et al. (Elife, 2018). These references are included in the review but should be indicated again in this sentence.

Author Response

This reviewer was likewise supportive of this piece and has mentioned that “The author provides a comprehensive view of the pioneer literature describing the main molecular complex at the core of spindle orientation...Another interesting point of this review is that the author considers not only the molecular regulation of the ternary complex, but also its interaction with the actin cytoskeleton, and its modulation by external forces”. However, also invited me to address some of her/his concerns and how this has been achieved is explained below.

Major Concerns:

1.         It would be important that the author justifies the choice of focusing on C.elegans and cultured cells, mentioning, for instance, the advantages of the models and/or the resemblances between them. Also, the author should mention briefly other main models in which the core regulators were first described (Drosophila neuroblasts, for instance).

Response. I thank the reviewer for his/her generous remarks and insightful suggestions. The reviewer will see now in the introduction on p. 4, I have explicitly mentioned that I will be focusing on C. elegans and cultured mammalian cells. As also suggested by the first reviewer (see the related point #1), I have also modified the title to emphasize more on the content of the review. Further, I have provided an exhaustive list of review articles covering polarity/spindle positioning in various systems such as Drosophila neuroblasts, Drosophila sensory organ progenitor cells (SOPs), C. elegans embryos, and mammalian epithelial cells on p. 4. I have focused on C. elegans embryos and mammalian cells in culture, firstly because of my expertise, and secondly the advantages of using evolutionarily divergent systems to address the conserved/non-conserved pathways of spindle positioning. However, as the reviewer suggested I have not attempted to justify this in a review merely because my justification may reflect some biases from my side that I like to avoid if that is alright by the reviewer.

2.         There is a lack of description of the different models of spindle orientation or positioning in cultured cells: spindle orientation concerning the substratum/spindle orientation guided by micropatterns/spindle positioning along its own axis. The mechanisms acting in each type of orientation might be different, so the author should describe better these different models, and specify throughout the review in which model each mechanism was found. Similarly, the explanation of the micropattern based system is incomplete (lines 128-129).

Response.    I wholeheartedly agree with the reviewer that there could be molecular differences in studying the spindle positioning with respect to substratum, spindle positioning guided by the micro-patterns and spindle positioning along its own axis. As the reviewer will notice now that I have tried to fill this gap by providing this information whenever required for instance on p. 6. I have also included an explanation of the micro-patterns-based system on p. 6 and 9.

3.         The author could go deeper in the discussion on the interaction of the ternary complex with actin-related proteins and the actin cortex, by mentioning more example proteins or other mechanisms by which the actin cortex may regulate spindle orientation (e.g. Machicoane et al., JCB 2014; Castanon et al., NCB 2013). What do we know about the contribution of the actomyosin cytoskeleton to spindle positioning in C.elegans? (e.g:  Redemann et al., Plos One 2010, Sugioka and Bowerman, Dev Cell 2018).

Response.  As requested by the reviewer I have now extended the discussion on the interaction of the ternary complex components with the actin-related proteins by giving more examples as suggested by the reviewer, and I thank the reviewer for the same. Also, as mentioned by the first reviewer, I have now extended a discussion on the role of actomyosin on spindle positioning in the section ‘Linking extrinsic mechanical forces to spindle positioning’ on p.11’ see the related major point (#3) of the first reviewer.  Castanon et al., NCB 2013 have revealed the importance of Anthrax toxin receptor 2a and actin cytoskeleton in regulating spindle along the A-V axis in Zebrafish, I have decided not to include this as the focus is mainly C. elegans and cultured cells, I hope this is OK with the reviewer.

4.         In lines 195 to 198, the author suggests a link between the ternary complex, cellular junctions and actin, and its function in tissues. There are novel insights into this subject, so some references should be at least included (e. g. Bosveld et al., Nature 2016).

Response.  In the earlier version of the review article, I did not include Bosveld et al., Nature 2016 since the focus was mainly on worms and mammalian cells as a model. However, as pointed out by the reviewer there are novel insights and similarities between Drosophila epithelia and mammalian cells (please see p. 7), and I have included this reference in the revised manuscript, Thanks.

5.    In the Section « Coupling extrinsic mechanical forces to instructing spindle positioning »:

While the author indicates that the link between retraction fibers, actin clouds and the ternary complex is not clear (lines 220 to 222), the articles by Machicoane et al. (JCB, 2014) and Tame et al. (Cell Cycle, 2014) link the localization of the ternary complex with the distribution of retraction fibers. Besides, the article by Kwon et al. (Dev Cell, 2015) strongly suggests that the LGN-dynein pathway acts in parallel to the actin cloud-myosin 10 pathway.

Response.  I further thank the reviewer for these suggestions! I have now improved the link between retraction fibres, actin cloud and the ternary complex on p. 9-11. The reviewer will further notice that I have added the references of Machicoane et al. (JCB, 2014) and Tame et al. (Cell Cycle, 2014) and also mentioned that LGN-dynein pathway acts in parallel to the actin cloud-myosin pathway on p. 10-11.

6.         In section « Spatiotemporal control of spindle positioning »:

-Plk1 was suggested to regulate both dynein and LGN cortical localization in separate studies (Kiyomitsu and Cheeseman, NCB 2012; Tame et al., EMBO reports, 2016); the author could discuss both datasets.

Response.  As suggested by the reviewer I have now extended the discussion on the role of Plk1 in spindle positioning by Kiyomitsu and Cheeseman, NCB 2012; Tame et al., EMBO reports, 2016, and I have cited them individually. Also, I have further discussed a new publication by our group on how Plk1 regulate spindle positioning by controlling the localization of the ternary complex components (Sana et al., 2018 Life Science Alliance)

            -A temporal description of the biochemical/localization events differing between metaphase and anaphase is lacking, in particular with respect to NuMA phosphorylation states and its cortical receptors (Kiyomitsu and Cheeseman, Cell 2013; Kotak et al., EMBO J 2013; Kotak et al., EMBO J 2014).

Response.  Thanks for these suggestions as well. I have now included the function of NuMA/dynein complex in anaphase, and also what is the molecular mechanisms of NuMA/dynein enrichment at the cell cortex in anaphase on p. 7. In addition to what is suggested by the reviewer, I have also cited the work from the laboratories of Terry Lechler, Qunasheng Du and Patricia Wadsworth in this theme. 

7.         The choice of dedicating a section to the role of Wnt in spindle orientation is unclear, as this section does not go deep enough in the subject and seems a bit disconnected to the rest of the review. I suggest to either remove this section, or extend its content.

Response.  I further thank the reviewer for her/his remarks. However, I am of the opinion that I should include what we currently know related to the role of Wnt signal on spindle positioning in worms and mammalian cells. As suggested by the reviewer I have now in greatly extended the discussion on the Wnt-based signalling in controlling spindle positioning in section ‘Extrinsic chemical code in guiding positioning of the mitotic spindle’ on page 12-13. There I have specifically tried to include all the past and present literature of how we view the positioning of the spindle is regulated in EMS cell. I have also included a new figure 3 to discuss, how spindle positioning is regulated in AB and EMS blastomere (p. 11 and 12-13 respectively). With these changes, I hope the reviewer will find this section coherent with the other parts.

Minor points:

1.    I suggest to better define spindle orientation and spindle positioning at the beginning of the review.

Response. Both terms, spindle positioning, and spindle orientation are invariably used in mammalian cells to define the spindle axis with respect to the substrate, and for the logical reason as C. elegans embryos are not round, spindle positioning is the favorable term in C. elegans literature. Throughout this review article, I stick with the term ‘spindle positioning’ to define the axis of the mitotic spindle in the 3-D cellular space.

2.      In lines 67-68 the author states that all spindle orientation mechanisms rely on astral microtubules. There are some exceptions as shown by Lazaro Dieguez et al. (Mol Biol of the Cell, 2015).

Response.    Thanks for suggesting this, I have now modified the sentence by stating that in ‘most of these mechanisms primarily rely on the dynamic astral microtubules that emanate from the centrosomes’ on p. 3. I hope the reviewer concur with me on this.

3.         The author mentions that the regulation of the spindle orientation ensures « error-free » cell division, or « flawless » division. However, spindle orientation per se does not ensure error free division neither spindle misorientation induces defective mitosis (concerning genomic integrity at the end of division, if this is what the author meant).

Response. I fully agree with the reviewer that in C. elegans zygote and mammalian cells spindle positioning defects do not induce defective mitosis. However, in anaphase, it is suggested that the inability of a cell of not having proper spindle elongation can impact genomic integrity. Since mis-positioning of the mitotic spindle is a sort of ‘error/flaw’ in contrast to the control cells where mitotic spindle aligns in a particular axis, I have utilized the term error-free/flawless. I have used these terms in the past in several publications, and I am happy to remove this if the reviewer thinks otherwise.

4.         In line 189, a reference to Kiyomitsu and Cheeseman (Cell, 2013) should be included.

Response. Thanks, added.

5.         In conclusion section: In line 342, a reference to articles using optogenetics to study spindle orientation should be included: Fielmich et al. (Elife, 2018) and Okumura et al. (Elife, 2018). These references are included in the review but should be indicated again in this sentence.

Response. Added, thanks!

Round 2

Reviewer 1 Report

The author has substantially improved this review, providing more coverage of other model systems outside of C. elegans, and more extensive coverage of the actomyosin cytoskeleton. These changes make this a very nice review that I think will be very useful for updating readers on the state of this field. I think the review can be published with no further revisions needed. 

Reviewer 3 Report

Dr Kotak provided a new version of his review article now entitled : « Mechanisms of spindle positioning : lessons from worms and mammalian cells ». Overall, the manuscript is improved with respect to the first version, and goes further in discussing the recent literature. The author included most of the suggestions proposed and addressed the different points requested.

I have a few more suggestions related to the points raised before, which should be reconsidered in the new version before publication:

1) Concerning my request of justifying the choice of C.elegans and mammalian cells for the review, the author replied that one possible argument are «  the advantages of using evolutionarily divergent systems to address the conserved/non-conserved pathways of spindle positioning »  . I suggest to include this idea in the review.

3) The author describes spindle positioning models in cultured cells, as follows : For instance, when HeLa cells are cultured on the uniform fibronectin substrate, spindle

align parallel to the substrate, however when such cells are cultured on the L-shape

fibronectin-based micro-patterns, spindle align in the longest axis […] .

It should be noted that in the micropattern based model, cells still orient their spindle in parallel to the substratum and in addition they show a second spindle orientation with respect to the shape of the pattern. The sentence should be rephrased to make this clear.

3)  In the section « Spatiotemporal control of spindle positioning », I suggest to mention the changes in NuMA phosphorylation between metaphase and anaphase, given that those changes directly influence NuMA interaction with alternative receptors and thus the cortical enrichment of NuMA in HeLa cells. This is relevant for this section. The author mentioned the anaphase receptor PIP2 in page 7 but not the relation with NuMa phosphorylation status.

4) Concerning the following exchange with the author cited below :

>    The author mentions that the regulation of the spindle orientation ensures « error-free » cell division, or « flawless » division. However, spindle orientation per se does not ensure error free division neither spindle misorientation induces defective mitosis (concerning genomic integrity at the end of division, if this is what the author meant).

Response. I fully agree with the reviewer that in C. elegans zygote and mammalian cells spindle positioning defects do not induce defective mitosis. However, in anaphase, it is suggested that the inability of a cell of not having proper spindle elongation can impact genomic integrity. Since mis-positioning of the mitotic spindle is a sort of ‘error/flaw’ in contrast to the control cells where mitotic spindle aligns in a particular axis, I have utilized the term error-free/flawless. I have used these terms in the past in several publications, and I am happy to remove this if the reviewer thinks otherwise.   <

I prefer the author to remove the terms « error-free/flawless » mitosis as the reader can be confused on what is the function of spindle positioning. Spindle elongation is still something different than spindle positioning so the link with genome integrity is not direct.